# Structural Pruning of Pre-Trained Language Models via Neural Architecture Search

## Abstract

Pre-Trained language models (PLM) mark the state-of-the-art for natural language understanding. However, their large size poses challenges in deploying them for inference in real-world applications, due to significant GPU memory requirements and high inference latency. This paper explores weight-sharing based neural architecture search (NAS) as a form of structural pruning to find sub-parts of the fine-tuned network that optimally trade-off efficiency, for example in terms of model size or latency, and generalization performance. Unlike traditional pruning methods with fixed thresholds, we propose to adopt a multi-objective approach that identifies the Pareto optimal set of sub-networks, allowing for a more flexible and automated compression process. Our NAS approach achieves up to $50\%$ compression with less than $5\%$ performance drop for a fine-tuned BERT model on 7 out of 8 text classification tasks.

## 1 Introduction

Pre-trained language models (PLMs) represent the current state-of-the-art for natural language understanding (NLU) tasks (Devlin et al., 2019). However, deploying PLMs for inference can be challenging due to their large parameter count. Current PLMs demand significant GPU memory and exhibit high inference latency, making them impractical for many real-world applications, for example when used in an end-point for a web service or deployed on an embedded systems. Recent work (Blalock et al., 2020; Kwon et al., 2022; Michel et al., 2019; Sajjad et al., 2022) demonstrated that in many cases only a subset of the pre-trained model significantly contributes to the downstream task performance. This allows for compressing the model by pruning parts of the network while minimizing performance deterioration.

Unstructured pruning (Blalock et al., 2020) computes a score for each weight in the network, such as the weight's magnitude, and removes weights with scores below a predetermined threshold. This approach often achieves high pruning rates with minimal performance degradation, but it also leads to sparse weight matrices, which are not well-supported by commonly used machine learning frameworks. Structured pruning (Michel et al., 2019; Sajjad et al., 2022) removes larger components of the networks, such as layers or heads. Although it typically does not achieve the same pruning rates as unstructured pruning, it only prunes entire columns/rows of the weight matrix, making it compatible with popular deep learning frameworks and hardware.

Recent work on neural architecture search (Zoph & Le, 2017; Real et al., 2017; Bergstra et al., 2013) (NAS) finds more resource efficient neural network architectures in a data-driven way. To reduce the computational burden of vanilla NAS, weight-sharing-based neural architecture search (Pham et al., 2018; Liu et al., 2019b; Elsken et al., 2018) first trains a single *super-network* and than searches for *sub-networks* within the super-network. It can be considered as a form of structural pruning, where one aims to find sub-networks that sustain performance of the given super-network. Most structural pruning approaches prune the networks based on a predefined threshold on the pruning ratio. In scenarios where there is no strict constraint on model size, it can be challenging to define such a fixed threshold in advance. NAS offers a distinct advantage over other pruning strategies by enabling a *multi-objective approach* to identify the Pareto optimal set of sub-networks, which captures the *nonlinear relationship* (see Figure 1) between model size and performance instead of just obtaining a single solution. This allows us to automate the compression process and to select the

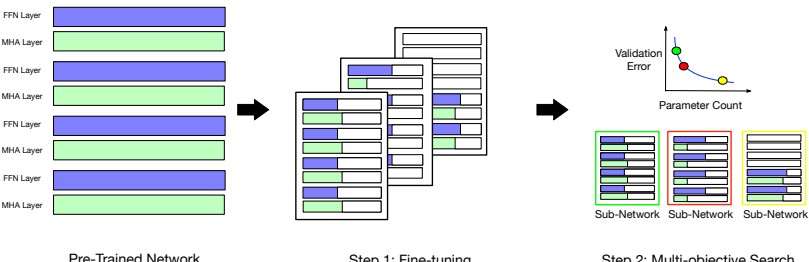

Figure 1: Illustration of our approach. We fine-tune the pre-trained architecture by updating only sub-networks, which we select by placing a binary mask over heads and units in each MHA and FFN layer. Afterwards, we run a multi-objective search to select the optimal set of sub-networks that balance parameter count and validation error.

best model that meets our requirements post-hoc after observing the non-linear Pareto front, instead of running the pruning process multiple rounds to find the right threshold parameter.

While there is a considerable literature on improving the efficiency of LLM, to the best of our knowledge there is no work yet that explored the potential of NAS for pruning fine-tuned PLMs. Our contributions are the following:

- We discuss the intricate relationship between weight-sharing based NAS and structural pruning and present a NAS approach that compresses PLMs for inference after fine-tuning on downstream tasks, while minimizing performance deterioration. Our focus lies *not* in proposing a novel NAS method per se, but rather in offering a *practical use-case* for NAS in the context of LLM.

- We propose four different search spaces to prune components of transformer based LLM and discuss their complexity and how they affect the structure of sub-networks. We also show how existing structural pruning approaches operate in two of these search space.

- Our method offers a more accurate approximation of the Pareto front that better balances generalization performance and parameter count than running state-of-the-art structural pruning techniques multiple times with different thresholds.

- We perform a thorough ablation study of weight-sharing based NAS and show that this use case serves as a useful test bed to benchmark NAS methods. In the long run we anticipate that our work will drive the development of future NAS methods.

We present an overview of related work in Section 2 and describe our methodology in Section 3. Section 4 provides an empirical comparison of our proposed approach with other structural pruning methods from the literature, along with an in-depth ablation study.

## 2 RELATED WORK

**Neural Architecture Search** (NAS) (see Elsken et al. (2018) for an overview) automates the design of neural network architectures to maximize generalization performance and efficiency (e.g., in terms of latency, model size or memory consumption). The limiting factor of convential NAS is the computational burden of the search, which requires multiple rounds of training and validating neural network architectures (Zoph & Le, 2017; Real et al., 2017). To mitigate this cost, various approaches have been proposed to accelerate the search process. For example, some of these methods early terminate the training process for poorly performing configurations (Li et al., 2018) or extrapolating learning curves (White et al., 2021b). Weight-sharing NAS (Pham et al., 2018; Liu et al., 2019a) addresses the cost issue by training a single super-network consisting of all architectures in the search space, such that each path represent a unique architecture. Initially, Liu et al. (2019a) framed this as a bi-level optimization problem, where the inner objective involves the optimization of the network weights, and the outer objective the selection of the architecture. After training the super-network, the best architecture is selected based on the shared weights and then re-trained from scratch. However, several papers (Li & Talwalkar, 2020; Yang et al., 2020) reported

that this formulation heavily relies on the search space and does not yield better results than just randomly sampling architectures. To address this limitation, Yu et al. (2020) proposed a two-stage NAS process. In the first stage, the super-network is trained by updating individual sub-networks in each iteration, instead of updating the entire super-network. After training, the final model is selected by performing gradient-free optimization based on the shared weights of the super-network, without any further training. Concurrently, Cai et al. (2020) applies a similar approach for convolutional neural networks in the multi-objective setting by first training a single super-network and then searching for sub-networks to minimize latency on some target devices. Related to our work is also the work by Xu et al. (2021), which searches for more efficient BERT architectures during the pre-training phase.

**Structural Pruning** involves removing parts of a trained neural network, such as heads (Michel et al., 2019), or entire layers (Sajjad et al., 2022), to reduce the overall number of parameters while preserving performance. Individual components are pruned based on a specific scoring function, using a manually defined threshold. For transformer-based architectures, Michel et al. (2019) observed that a significant number of heads, up to a single head in a multi-head attention layer, can be deleted after fine-tuning without causing a significant loss in performance. Voita et al. (2019) proposed L0 regularization as a means to prune individual heads in a multi-head attention layer. Kwon et al. (2022) prunes individual heads and units in the fully-connected layers after fine-tuning according to the Fisher information matrix. Sajjad et al. (2022) demonstrated that it is even possible to remove entire layers of a pre-trained network prior to fine-tuning, with minimal impact on performance. In comparison to our data-driven approach, Sajjad et al. (2022) suggested using predefined heuristics (e.g., deleting top / odd / even layers) to determine layers to prune. However, as shown in our experiments, the appropriate architecture depends on the specific task, and more data-driven methods are necessary to accurately identify the best layers to prune.

**Distillation** (Hinton et al., 2015) trains a smaller student model to mimic the predictions of a pre-trained teacher model. For instance, Sanh et al. (2020) used this approach to distill a pre-trained BERT model (Devlin et al., 2019) into a smaller model for fine-tuning. Jiao et al. (2019) proposed a knowledge distillation approach specifically for transformer-based models, which first distills from a pre-trained teacher into a smaller model and then performs task-specific distillation in a second step based on a task augmented dataset. Related to our method is also AdaBERT (Chen et al., 2020) which trains task-specific convolutional neural networks based on differentiable NAS (Liu et al., 2019a) by distilling the knowledge of a PTL such as BERT. Unlike pruning-based methods, distillation allows for complete architectural changes beyond merely dropping individual components. However, from a practical standpoint, determining the optimal structure and capacity of the student network needed to match the performance of the teacher network also amounts to a hyperparameter and neural architecture search problem. Additionally, training a student network requires a significant amount of computational resources. For example, Sanh et al. (2020) was trained for around 90 hours on 8 16GB V100 GPUs. This cost can be amortized by fine-tuning the student model to solve many different tasks but depending on the downstream tasks, it potentially requires a substantial amount of iterations which is not always desirable for practitioners who aim to solve a single specific task. This is especially important in the multi-objective setting where many networks need to be distilled to map the full size/accuracy Pareto front.

**Quantization** (Dettmers et al., 2022; Dettmers & Zettlemoyer, 2023) reduces the precision of model parameters from floating-point numbers to lower bit representations (e.g., 8-bit integers). The main advantage of quantization is the reduction in memory footprint. However, as we show in the Appendix E, this does not necessarily lead to faster latency. Quantization is independent of our NAS approach and can be employed on the pruned network to further decrease memory usage.

## 3 STRUCTURAL PRUNING VIA NEURAL ARCHITECTURE SEARCH

We first provide a multi-objective problem definition for compressing fine-tuned LLM. Afterwards, we describe our weight-sharing based NAS approach and present four search spaces to prune transformer-based architectures, with a different degree of pruning.

### 3.1 PROBLEM DEFINITION

We consider a pre-trained transformer model based on an encoder-only or decoder-only architecture, such as for example BERT (Vaswani et al., 2017), with $L$ non-embedding layers, each composed of a multi-head attention (MHA) layer followed by a fully connected feed forward (FFN) layer. Given an input sequence $\boldsymbol{X} \in \mathbb{R}^{n \times d_{model}}$, where $n$ represents the sequence length and $d_{model}$ the size of the token embedding, the MHA layer is defined by: $MHA(\boldsymbol{X}) = \sum_i^H Att(\boldsymbol{W}_Q^{(i)}, \boldsymbol{W}_K^{(i)}, \boldsymbol{W}_V^{(i)}, \boldsymbol{W}_O^{(i)}, \boldsymbol{X})$ where $\boldsymbol{W}_Q^{(i)}, \boldsymbol{W}_K^{(i)}, \boldsymbol{W}_V^{(i)} \in \mathbb{R}^{d_{model} \times d}$ and $\boldsymbol{W}_O^{(i)} \in \mathbb{R}^{Hd \times d_{model}}$ are weight matrices. $Att(\cdot)$ is a dot product attention head (Bahdanau et al., 2015) and $H$ is the number of heads. The output is then computed by $\boldsymbol{X}_{MHA} = LN(\boldsymbol{X} + MHA(\boldsymbol{X}))$, where LN denotes layer normalization (Ba et al., 2016). The FFN layer is defined by $FFN(\boldsymbol{X}) = \boldsymbol{W}_1\sigma(\boldsymbol{W}_0\boldsymbol{X})$, with $\boldsymbol{W}_0 \in \mathbb{R}^{I \times d_{model}}$ and $\boldsymbol{W}_1 \in \mathbb{R}^{d_{model} \times I}$, where $I$ denotes the intermediate size and $\sigma(\cdot)$ is a non-linear activation function. Also here we use a residual connection to compute the final output: $x_{FFN} = LN(\boldsymbol{X}_{MHA} + FFN(\boldsymbol{X}_{MHA}))$.

We define a binary mask $\boldsymbol{M}_{head} \in \{0,1\}^{L \times H}$ for each head in the multi-head attention layer and a binary mask $\boldsymbol{M}_{neuron} \in \{0,1\}^{L \times U}$ for each neuron in the fully-connected layers. The output of the $l$-th MHA layer and FFN layer is computed by $MHA_l(\boldsymbol{X}) = \sum_i^H \boldsymbol{M}_{head}[i,l] \circ Att(\cdot)$ and $FFN_l(\boldsymbol{X}) = \boldsymbol{M}_{neuron}[l] \circ W_1\sigma(W_0\boldsymbol{X})$, respectively.

Now, let's define a search space $\boldsymbol{\theta} \in \Theta$ that contains a finite set of configurations to define possible sub-networks sliced from the pre-trained network. We define a function CREATEMASK that maps from a configuration $\boldsymbol{\theta} \rightarrow \boldsymbol{M}_{head}, \boldsymbol{M}_{neuron}$ to binary masks. Let's denote the function $f_0 : \Theta \rightarrow \mathbb{R}$ as the validation error of the sub-network defined by configuration $\boldsymbol{\theta}$ after fine-tuning on some downstream task. To compute the validation score induced by $\boldsymbol{\theta}$ we place corresponding masks $\boldsymbol{M}_{head}, \boldsymbol{M}_{neuron}$ over the network. Additionally, we define the total number of trainable parameter $f_1 : \Theta \rightarrow \mathbb{N}$ of the subnetwork. Our goal is to solve the following multi-objective optimisation problem:

$$min_{\boldsymbol{\theta} \in \Theta}(f_0(\boldsymbol{\theta}), f_1(\boldsymbol{\theta})). \tag{1}$$

In the multi-objective setting, there is no single $\boldsymbol{\theta}_\star \in \Theta$ that simultaneously optimizes all $M$ objectives. Let's define $\boldsymbol{\theta} \succ \boldsymbol{\theta}'$ iff $f_i(\boldsymbol{\theta}) \leq f_i(\boldsymbol{\theta}'), \forall i \in [M]$ and $\exists i \in [k] : f_i(\boldsymbol{\theta}) < f_i(\boldsymbol{\theta}')$. We aim to find the *Pareto Set*: $P_f = \{\boldsymbol{\theta} \in \Theta | \nexists \boldsymbol{\theta}' \in \Theta : \boldsymbol{\theta}' \succ \boldsymbol{\theta}\}$ of points that dominate all other points in the search space in at least one objective.

### 3.2 WEIGHT-SHARING BASED NAS

Following previous work (Yu et al., 2020; Wang et al., 2021), our weight-sharing based NAS approaches consists of two stages: the first stage is to treat the pre-trained model as super-network and fine-tune it on the downstream task, such that sub-networks do not co-adapt. The second stage, utilizes multi-objective search strategies to approximate the Pareto-optimal set of sub-networks (see Figure 1 for an illustration).

#### 3.2.1 SUPER-NETWORK TRAINING

In the standard NAS setting, we would evaluate $f_0(\boldsymbol{\theta})$ by first fine-tuning the sub-networks defined by $\boldsymbol{\theta}$ on the training data before computing the score on the validation data. The weights of the sub-network are initialize based on the pre-trained weights. While more recent NAS approaches (Li & Talwalkar, 2020; Klein et al., 2020) accelerate the search process by early stopping poorly performing sub-networks, this still amounts to an optimization process that requires the compute of multiple independent fine-tuning runs.

The idea of two-stage weight-sharing-based NAS (Yu et al., 2020) is to train a single-set of shared weights, dubbed super-network, that contains all possible networks in the search space. After training the super-networks, evaluation $f_0(\boldsymbol{\theta})$ only requires a single pass over the validation data.

We consider the pre-trained network as super-network with shared weights that contains all possible sub-networks $\boldsymbol{\theta} \in \Theta$. To avoid that sub-networks co-adapt and still work outside the super-network, previous work (Yu et al., 2020; Wang et al., 2021) suggested to update only a subset of sub-networks in each update step, instead of the full super-network. We adapt this strategy and sample sub-

networks according to the sandwich rule (Yu et al., 2020; Wang et al., 2021) in each update step, which always updates the smallest, the largest and $k$ random sub-networks. The smallest and largest sub-network correspond to the lower and upper bound of $\Theta$, respectively. For all search spaces $\Theta$ define below, the upper bound is equal to full network architecture, i.e, the super-network and the lower bound removes all layer except the embedding and classification layer.

Additionally, we use in-place knowledge distillation (Yu et al., 2019) which accelerate the training process of sub-networks. Given the logits $\pi_{supernet}(\boldsymbol{x})$ of the super-network, which we obtain for free with the sandwich rule, and the logits of a sub-network $\pi_{\boldsymbol{\theta}}(\boldsymbol{x})$, the loss function to obtain gradients for the sub-networks follows the idea of knowledge distillation:

$$\mathcal{L}_{KD} = \mathcal{L}_{CE} + D_{\mathrm{KL}}\left(\sigma(\frac{\pi_{supernet}}{T}), \sigma(\frac{\pi_{\boldsymbol{\theta}}}{T})\right), \tag{2}$$

where $D_{\mathrm{KL}}(\cdot)$ denotes the Kullback-Leibler divergence between the logits of the super-network and the sub-network, $T$ a temperature parameter, $\sigma(\cdot)$ the softmax function and $\mathcal{L}_{CE}$ is the cross-entropy loss.

### 3.2.2 SUB-NETWORKS SELECTION

After training the super-network, we compute the validation error $f_0(\boldsymbol{\theta})$ by applying $\boldsymbol{M}_{head}$ and $\boldsymbol{M}_{neuron}$ to the shared weights and performing a single pass over the validation data. This substantially reduces the computational cost involved in the multi-objective problem stated in Equation 1.

Previous work (White et al., 2021a) has demonstrated that simple local search often performs competitively compared to more advanced NAS methods. In this paper, we propose a straightforward multi-objective local search approach. Starting from the current Pareto front $P_f$, which is initialized by some starting point, we randomly sample an element $\boldsymbol{\theta}_\star \sim P_f$ and then generate a random neighbor point by permuting a single random entry of $\boldsymbol{\theta}_\star$. The pseudo code for our local search is provided in Appendix F.

### 3.3 SEARCH SPACE

The search space $\Theta$ defines sub-networks of the pre-trained network architecture. An expressive $\Theta$ allows for fine-grained pruning but might also become infeasible to explore. We propose the following search spaces that exhibit different levels of complexity. For each search space we provide pseudo code to define the CREATEMASK function in Appendix B.

- **LARGE**: For each head and neuron in the fully-connected layer we define a single binary $\Theta_i = \{0, 1\}$ which is combined to form the search space $\Theta = \Theta_0 \times \ldots \times \Theta_{L(H+I)}$. This is the most expressive search space, but also grows quickly with the model size. The search space is also commonly used by other structural pruning approaches (Kwon et al., 2022). It might not be very useful in practice, because we cannot easily remove single rows/columns of the weight matrix with most transformer implementations and hence it will not necessarily reduce the inference latency. However, it provides us a reference in terms of predictive performances that can be retained under a certain pruning ratio.

- **MEDIUM**: Based on the previous search space, we allow for a flexible number of heads / units per layer. For each layer $l \in [0, L]$, we define $\mathcal{H}_l = [0, H]$ and $\mathcal{U}_l = [0, U]$, such that the final search space is $\Theta = \mathcal{H}_0 \times \mathcal{U}_0 \ldots \mathcal{H}_L \times \mathcal{U}_L$. For each layer, we always keep the first $h \in \mathcal{H}$ heads and $u \in \mathcal{U}$ units, respectively, to enforce that CREATEMASK is a bijective mapping (see Appendix B).

- **LAYER**: Inspired by Sajjad et al. (2022), we prune individual attention and fully-connected layers instead of single heads and neurons. We define a search space $\Theta = \{0, 1\}^L$ that contains one binary hyperparameter for each layer that determines if the corresponding layer is removed.

- **SMALL**: We define the number of heads $\mathcal{H} = [0, H]$, the number of units $\mathcal{U} = [0, U]$ and the total number of layers $\mathcal{L} = [0, L]$, such that $\Theta = \mathcal{H} \times \mathcal{U} \times \mathcal{L}$. Compared to the other search spaces, the dimensionality of this search space with different model sizes, and only its upper bound increases. As for the MEDIUM search space we also keep the first heads and units in each layer.

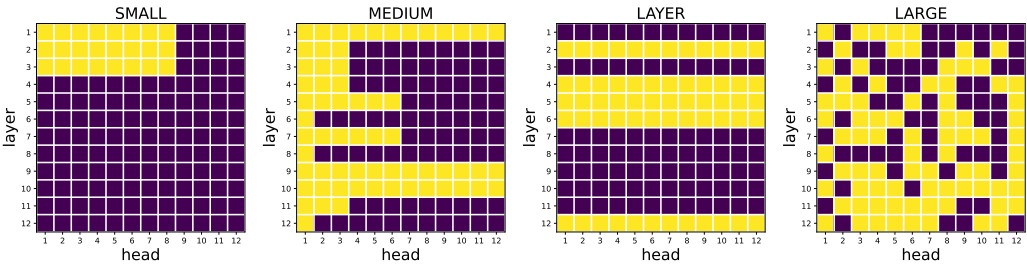

Figure 2: Examples of head masks $M_{head}$ sampled uniformly at random from different search spaces. Dark color indicates that the corresponding head is masked. The same pattern can be observed for $M_{neuron}$

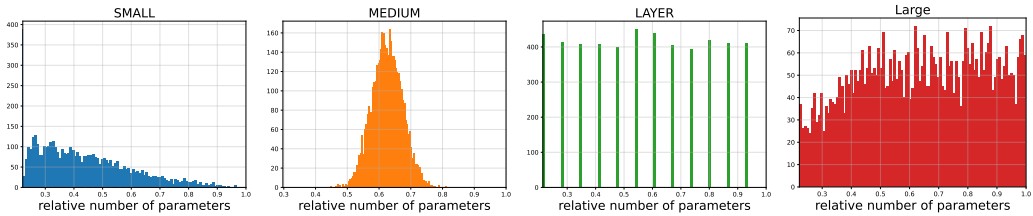

Figure 3: Distribution of the parameter count $f_1(\boldsymbol{\theta})$ for uniformly sampled $\boldsymbol{\theta} \sim \Theta$.

Each search space induces a different pattern for $M_{head}$ and $M_{neuron}$ that we place over the super-network to select sub-networks (see Figure 2 for some examples). To see how this effects the distribution over parameter count and hence the sampling during the super-network training, we sample $N = 500$ configurations $\{\theta_0, ..., \theta_N\}$ uniformly at random and compute the number of trainable parameters $\{f_1(\theta_i), ..., f_1(\theta_N)\}$ for all four search spaces (see Figure 3). The SMALL search space is somewhat biased to smaller networks. The MEDIUM search space, even though more expressive, is highly biased towards mid-size networks, since on average half of the heads / neurons are masked out. For the two binary search spaces LAYER and LARGE, we can achieve a uniform distribution over the number of parameters, by using the following sampling process. We first sample an integer $k \sim U(0, K)$, where $k = L$ for the LAYER search space, and $k = L(H + I)$ for the LARGE search space. Afterwards, we randomly select $k$ entries of the binary vector $\boldsymbol{\theta} \in \Theta$ and set them to 1.

## 4  EXPERIMENTS

We evaluate our approach on eight text classification datasets from the GLUE (Wang et al., 2019) benchmark suite. We provide a description of each dataset in Appendix C. All datasets come with a predefined training and evaluation set with labels and a hold-out test set without labels. We split the training set into a training and validation set ($70\%/30\%$ split) and use the evaluation set as test set. We fine-tune every network for 5 epochs on a single GPU. For all multi-objective search methods, we use Syne Tune (Salinas et al., 2022) on a single GPU instance. We use BERT-base (Devlin et al., 2019) (cased) as pre-trained network, which consists of $L = 12$ layers, $I = 3072$ units and $H = 12$ heads (other hyperparameters are described in Appendix A), because it achieved competitive performance too larger models on these benchmarks and allows for a more thorough evaluation. We also present a comparison to quantization in Appendix E.

### 4.1  COMPARISON

We now present a comparison against other structural pruning approaches. For NAS we use the SMALL search space defined in Section 3.3 based on our ablation study in Section 4.2. We compare against the following relevant baselines:

- **Retraining Free Pruning** (RFP) (Kwon et al., 2022) uses a three-phased pruning strategy that, based on a threshold $\alpha$, prunes individual heads in the MHA layer and units in the

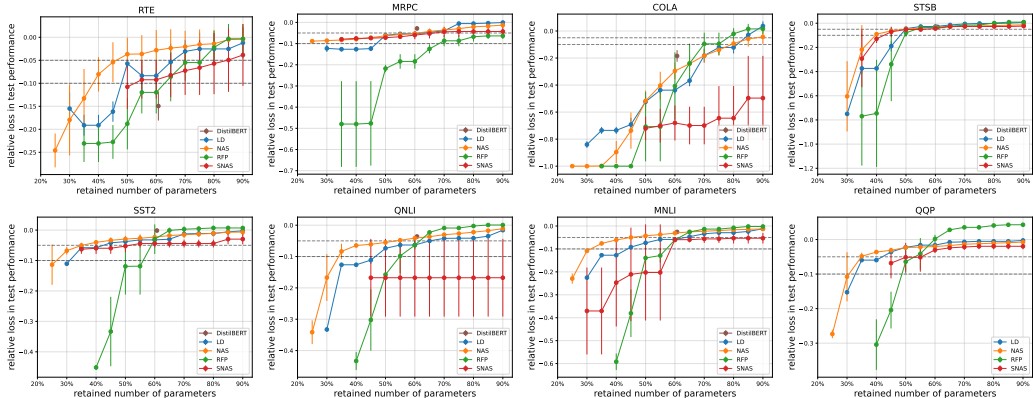

Figure 4: Loss in test performance versus the parameter count relative to the un-pruned BERT-base-cased model on all 8 text classification datasets. On 7 out of 8 dataset our NAS strategy is able to prune $50\%$ with less than $5\%$ drop in performance (indicated by the dashed line) in performance.

FFN layer. The first phase computes a binary mask for heads and units by computing the diagonal Fisher information matrix. The matrix is then rearranged by a block-approximated Fisher information matrix. In the last step, the masked is further tuned by minimizing the layer-wise reconstruction error. This method operates in the LARGE search space described in Section 3.3. We run RFP with different values for $\alpha \in \{0.1, 0.2, ..., 0.9\}$ to obtain a Pareto set of architectures.

- **Layer Dropping** (LD): Following Sajjad et al. (2022) we first remove the top $n \in 1, ..., L-1$ layers and fine-tune the remaining layers directly on the downstream task. To obtain a Pareto set of $N$ points, we fine-tune $N$ models with different amount of layers removed. This method serves as a simple heuristic to explore the LAYER search space.

- **DistilBERT** (Sanh et al., 2020) is a distilled version of BERT based on a smaller architecture ($L = 6, H = 12, I = 3072$) which we directly fine-tuned on the downstream task.

- **Standard NAS** (S-NAS) uses the same multi-objective search but without the super-network training. Instead each sub-network is initialized with the pre-trained weights and then fine-tuned independently.

For each method except DistilBERT, we obtain a Pareto set of solutions with different parameter counts; note that parameter count is related to model inference time as discussed in E. To compare results, we normalize the number of parameters to $[0, 1]$ and bin results based on different thresholds $\beta \in \{0.2, ...0.9\}$. Note that roughly $20\%$ of the parameters of BERT-base are included in the embedding and classification head, and hence cannot be pruned. For each bin, we report the best performance of the solution with $\leq \beta$ parameters.

Figure 4 shows the parameter count (horizontal axis) and the test error (vertical axis) relative to the unpruned network for all datasets. For reference, we indicate $5\%$ and $10\%$ relative error to the unpruned network by dashed lines. NAS achieves strong performance, especially for higher pruning ratios. For smaller pruning ratios, i.e larger parameter counts (right side of plots), all methods exhibit comparable performance. Notably, NAS showcases more fine-grained pruning capabilities compared to LD, as demonstrated by the smooth curves in the results.

Apart from the quality of the final Pareto set, we also evaluate the total runtime of each method. Figure 5 left shows the total runtime in terms of wall-clock time for the MNLI dataset. Plots for all other datasets are in Appendix D. For both, RFP and NAS, we also include the fine-tuning of the super-network in the runtime analysis. LD exhibits significantly higher runtime, in terms of wall-clock time compared to NAS, since it fine-tunes $n$ sub-networks. While RFP is overall faster, our NAS approach provides the best performance / runtime trade-off.

For a qualitative comparison, we show the results for a single run on the SST2 dataset in Figure 5 right. On this dataset our NAS approach finds sub-networks with approximately $50\%$ the size of the unpruned network (dashed line) with almost no drop in performance.

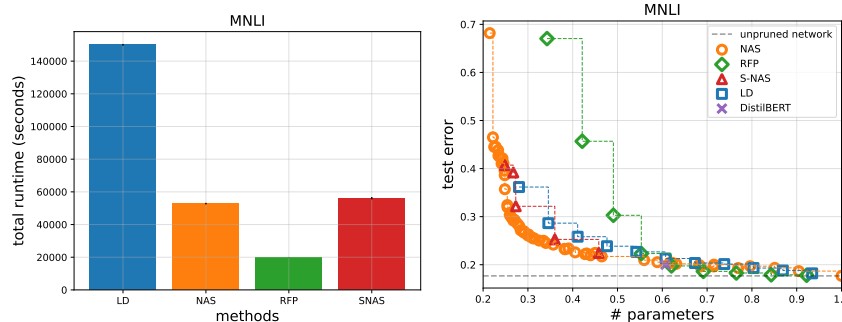

Figure 5: Total runtime in seconds for each method (left), including training time for the super-network, to generate the Pareto fronts (right) on the MNLI dataset. While RFP is faster than our NAS approach, its Pareto front performs poorly in the smaller sub-network regime. LD and S-NAS exhibit similar performance on this benchmark, but they consume substantially more resources.

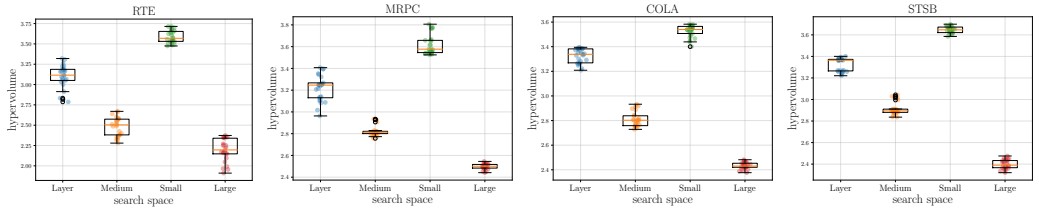

Figure 6: Comparison of different search spaces to define sub-networks. Even though larger search spaces are more expressive, they under-perform within the select budget.

## 4.2 ABLATION STUDY

We now present a detailed ablation study to evaluate different components of our NAS approach. To quantify the performance of a Pareto set, we compute the Hypervolume (Zitzler et al., 2003) frequently used in the multi-objective literature. We first normalize each objective based on all observed values across all methods and repetitions via Quantile normalization. This results in a uniform distribution between $[0, 1]$, and we use $(2, 2)$ as reference point. We train each super-network five times with a different random seed. For each model checkpoint, i.e super-network, we run multi-objective search five times also with different random seeds. This leads to 25 different Pareto sets and we report mean and total variance of the corresponding hypervolume. To cut computational cost, we report results only on the four smallest datasets: RTE, MRPC, COLA and STSB.

### 4.2.1 SEARCH SPACE

First, we compare the search spaces definitions from Section 3.3. We fine-tune the super-network as described in Section 3.2 and sample 100 sub-networks uniformly at random to compute the hypervolume. Within this budget (see Figure 6), the *SMALL* search space achieves the best performance. Interestingly, even though the *MEDIUM* search space allows for a more fine-grained per layer pruning, it leads to worse results. We attribute this to the non-uniform distribution of parameter count as described in Section 3.3. The *LARGE* search space, which is a superset of the other search spaces, seems infeasible to explore with random sampling over so few observations. We use the SMALL search space for the remaining experiments.

### 4.2.2 SUPER-NETWORK TRAINING

Next, we compare the following super-network training strategies:

- **standard**: Which trains all weights of super-network in the standard fine-tuning setting
- **random**: Samples a single random sub-network in each update steps

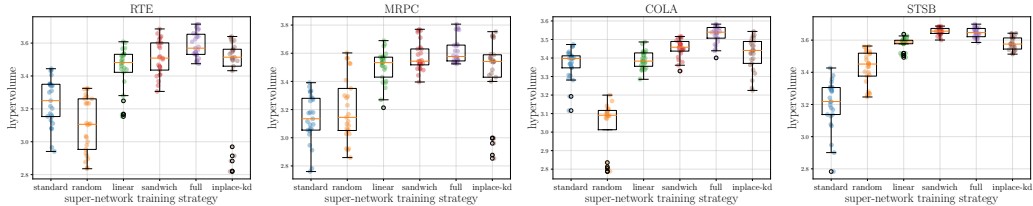

Figure 7: Comparison of super-network training strategies. More advances strategies that sample a set of sub-network outperform standard fine-tuning or just sampling a single random sub-network.

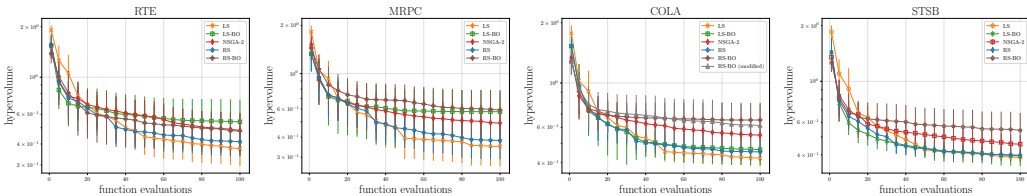

Figure 8: Hypervolume of different multi-objective search methods over the number of function evaluation. We report the difference to the optimal hypervolume given the reference point.

- **random-linear**: Following Yu et al. (2020), we either sample a random sub-network with probability $p$ or the full-network with probability of $1 - p$ in each update step. Thereby, $p$ is linearly increased from 0 to 1 after each update step over the course of training.

- **sandwich**: The super-network is updated according to the sandwich rule described in Section 3.2. We set the number of random sub-networks in each update step to $k = 2$.

- **kd**: Update $k = 2$ random sub-networks according to Equation 2.

- **full**: Implements the training protocol described in Section 3.2, i.e it combines the sandwich rule with in-place knowledge distillation to update sub-networks.

Figure 7 middle shows the hypervolume across all repetitions. Standard fine-tuning and just randomly sampling a sub-network leads to significant worse results. Linearly increasing the probability of sampling a random sub-networks stabalizes results. Better results are achieved by using the sandwich rule or knowledge distilation. Thereby, combining both slightly improves results further.

### 4.2.3 MULTI-OBJECTIVE SEARCH

Lastly, we compare in Figure 8 the following multi-objective search methods: our local search (LS) described in Section 3.2 (see Appendix F for details). Random search (RS) (Bergstra & Bengio, 2012) samples architectures uniformly at random from the search space. NSGA-2 (Deb et al., 2002) is a frequently used genetic algorithm from the multi-objective literature. Bayesian optimization (Garnett, 2023) with a linearized scalarization of the objectives (LS-BO) and with a randomizes scalarization of the objectives (RS-BO) (Paria et al., 2019).

While all methods yield comparable results (note the high uncertainty bars), LS performs slightly better. RS-BO and LS-BO under perform to RS because their scalarization approach causes them to concentrate solely on specific parts of the Pareto front, thereby failing to adequately capture its complete extent. NSGA-2 appears to suffer from sample inefficiency on this benchmark.

### 4.3 CONCLUSIONS

We propose weight-sharing-based NAS to compress fine-tuned PLMs by slicing sub-networks. By utilising a multi-objective approach, we can find the Pareto optimal set of architectures that balance model size and validation error, allowing practitioners to select the optimal network without running the pruning process multiple times with different thresholds. Furthermore, our method is more runtime efficient than baselines and more effective than structural pruning methods.

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

## A  HYPERPARAMETERS

Table A shows the hyperparameters for fine-tuning the super-network. We largely follow default hyperparameters recommended by the HuggingFace transformers library. For all multi-objective search method, we follow the default hyperparameter of Syne Tune.

| Hyperparameter | Value |
|---|---|
| Learning Rate | 0.00002 |
| Number of random sub-networks $k$ | 2 |
| Temperature $T$ | 10 |
| Batch Size | 4 |

## B  MASKING

Algorithm 1, 2, 3 and 4 show pseudo code for the LAYER, SMALL, MEDIUM and LARGE search space, respectively. Note that, $\mathbf{1}$ indicates a vector of ones. For a matrix $\boldsymbol{M}$, we write $\boldsymbol{M}[:, : N]$ to denote the first $N$ columns for all rows and, vice versa, $\boldsymbol{M}[: N, :]$ for the first $N$ rows.

**input**  : sub-network configuration $\boldsymbol{\theta} \in \{0, 1\}^L$
**output:** $\boldsymbol{M}_{head}, \boldsymbol{M}_{neuron}$
$\boldsymbol{M}_{head} \leftarrow [0]^{L \times H}$;
$\boldsymbol{M}_{neuron} \leftarrow [0]^{L \times I}$;
**for** $l = 0, \ldots, L - 1$ **do**
  $\quad \boldsymbol{M}_{head}[l, :] \leftarrow \boldsymbol{\theta}[l]$;
  $\quad \boldsymbol{M}_{neuron}[l, :] \leftarrow \boldsymbol{\theta}[l]$;
**end**

**Algorithm 1:** CREATEMASK function for LAYER search space

**input**  : sub-network configuration $\boldsymbol{\theta} \in \mathcal{H}_0 \times \mathcal{U}_0 \ldots \mathcal{H}_L \times \mathcal{U}_L$
**output:** $\boldsymbol{M}_{head}, \boldsymbol{M}_{neuron}$
$\boldsymbol{M}_{head} \leftarrow [0]^{L \times H}$;
$\boldsymbol{M}_{neuron} \leftarrow [0]^{L \times I}$;
**for** $l = 0, \ldots, L - 1$ **do**
  $\quad h = \boldsymbol{\theta}[2 * l]$ ;                    /* number of heads in layer $l$ */
  $\quad u = \boldsymbol{\theta}[2 * l + 1]$ ;                /* number of units in layer $l$ */
  $\quad \boldsymbol{M}_{head}[l, : h] \leftarrow \mathbf{1}$;
  $\quad \boldsymbol{M}_{neuron}[l, : u] \leftarrow \mathbf{1}$;
**end**

**Algorithm 2:** CREATEMASK function for MEDIUM search space

**input**  : sub-network configuration $\boldsymbol{\theta} \in \mathcal{H} \times \mathcal{U} \times \mathcal{L}$
**output:** $\boldsymbol{M}_{head}, \boldsymbol{M}_{neuron}$
$h = \boldsymbol{\theta}[0]$ ;                               /* number of heads */
$u = \boldsymbol{\theta}[1]$ ;                               /* number of units */
$l = \boldsymbol{\theta}[2]$ ;                               /* number of layers */
$\boldsymbol{M}_{head} \leftarrow [0]^{L \times H}$;
$\boldsymbol{M}_{neuron} \leftarrow [0]^{L \times I}$;
$\boldsymbol{M}_{head}[: l, : h] \leftarrow \mathbf{1}$;
$\boldsymbol{M}_{neuron}[: l, : u] \leftarrow \mathbf{1}$;

**Algorithm 3:** CREATEMASK function for SMALL search space

**input** : sub-network configuration $\boldsymbol{\theta} \in \{0, 1\}^{L*(H+U)}$
**output**: $\boldsymbol{M}_{head}, \boldsymbol{M}_{neuron}$
$\boldsymbol{M}_{head} \leftarrow \boldsymbol{\theta}[:, : H]$;
$\boldsymbol{M}_{neuron} \leftarrow \boldsymbol{\theta}[:, H :]$;

**Algorithm 4:** CREATEMASK function for LARGE search space

## C  DATASETS

We use the following 8 dataset from the GLUE (Wang et al., 2019) benchmarking library. All dataset are classification task, except for STSB, which is a regression dataset.

- The Recognizing Textual Entailment (RTE) dataset aims to identify the textual entailment of two sentences.

- The Microsoft Research Paraphrase Corpus (MRPC) dataset consists of sentence pairs extracted from online news sources. The task is to predicts if these pairs are semantically equivalent to each other.

- The Corpus of Linguistics Acceptability (COLA) dataset contains English sentences that are labeled as grammatically correct or not.

- The Semantic Textual Similarity Benchmark (STSB) consists of sentences pairs that are scored between 1 and 5 based on their similarity.

- The Stanford Sentiment Treebank (SST2) datasets classifies the positive / negative sentiment of sentences extracted from movie reviews.

- The Multi-Genre Natural Language Inference Corpus (MNLI) is a dataset with sentence pairs where one sentence represents a premise and the other sentence a hypothesis. The task is to predict whether the premise entails the hypothesis.

- QNLI is a modified version of the Stanford Question Answering Dataset which is a collection of question / answer pairs where question are written by human annotators and answers are extracted from Wikipedia. The task is to predict whether the answers is correct.

- Quora Question Pairs (QQP) dataset includes question pairs from the Quora website. The task is to predict whether two questions are semantically equivalent.

## D  ADDITIONAL RESULTS

In this section we present additional results that from the experiments described in Section 4. Figure 9 shows the mean standard deviation of the runtime for each method for all 8 datasets. For a detailed discussion see Section 4 in the main paper.

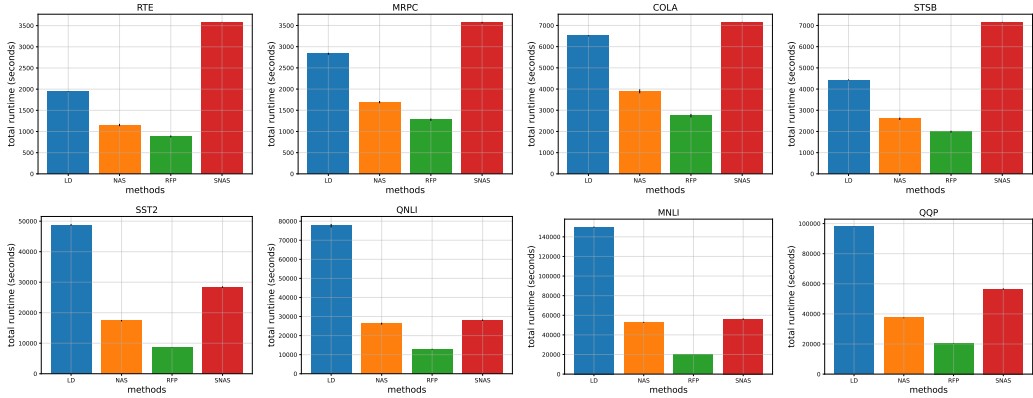

Figure 9: Runtime distributions for each method on all 8 GLUE datasets.

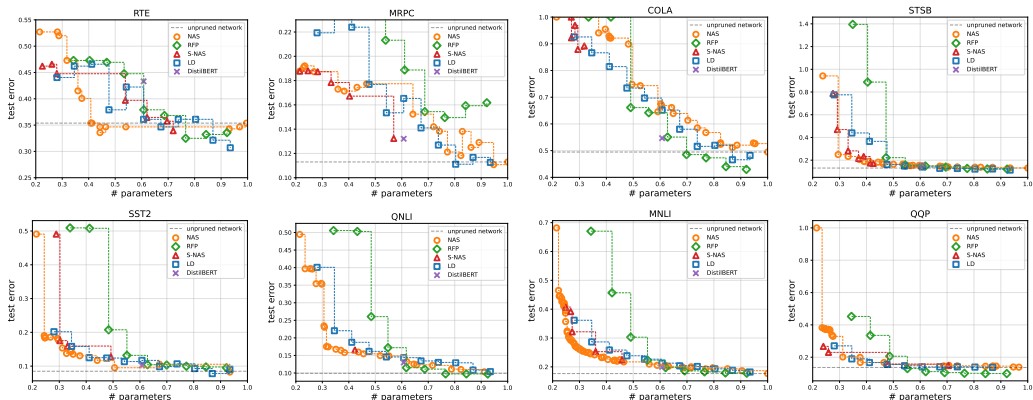

Figure 10: Single Pareto fronts of random run for each method on all 8 GLUE datasets.

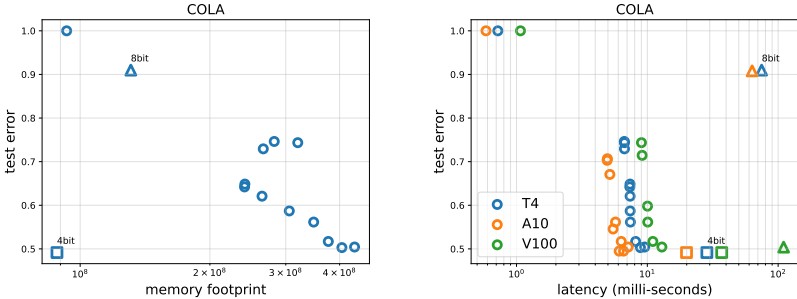

Figure 11: Test error versus memory footprint (left) and latency (right) on 3 different GPU types for the Pareto front found by our NAS strategy and the un-pruned network with 8bit and 4bit quantization.

# E  QUANTIZATION

Quantization (Dettmers et al., 2022; Dettmers & Zettlemoyer, 2023) is a powerful technique that significantly reduces the memory footprint of neural networks. However, its impact on latency is not immediate, especially when dealing with batch sizes that can not fit into the cache of the device Dettmers & Zettlemoyer (2023). With our flexible NAS framework we can simply replace objectives and directly optimize latency on the target device instead of parameter count.

Figure 11 left shows the Pareto set obtained with our NAS approach, where we optimize latency instead of parameter count on the COLA dataset across 3 different GPU types. Additionally, we evaluate the performance of the unpruned super-network with 8-bit (Dettmers et al., 2022) and 4-bit (Dettmers & Zettlemoyer, 2023) quantization. While quantization substantially reduces the memory footprint (Figure 11 right), it actually leads to worse latency. While quantization introduces a small overhead due to the additional rounding steps, the latency could potentially be reduced by optimizing the low-level CUDA implementation. Somewhat surprisingly using a int-8bit quantization leads to high performance drop on some hardware. NAS effectively reduces the sizes of weight matrices, leading to reduced GPU computation and, thus, is less hardware depend.

We can also apply quantization to sub-networks, making it orthogonal to our NAS methodology and offering further improvements to the memory footprint. Overall, these findings shed light on the trade-offs between memory footprint reduction and latency optimization. We leave it to future work to explore the connection between NAS and quantization.

# F    MULTI-OBJECTIVE LOCAL SEARCH

We start with evaluating a starting point $\boldsymbol{\theta}_{start}$, which we set as the upper bound of our search space. The initial Pareto front $P_0$ is initialized with the start point $P_0 \leftarrow \{\boldsymbol{\theta}_{start}\}$. Afterwards, in each step our local search samples a random neighbour of a randomly selected points of the current Pareto front until we reach a fix number of iteration. We consider 1-step neighbourhood, which randomly permutes the given point only in a single dimension.

**input** : Search space $\Theta$, number of iteration $T$, starting point $\boldsymbol{\theta}_{start}$
**output:** Pareto front $P$
```
/* evaluate starting point                                    */
```
$P_0 \leftarrow \{\boldsymbol{\theta}_{start}\}$;
$y_{start} = [f_0(\boldsymbol{\theta}_{start}), f_1(\boldsymbol{\theta}_{start}]$;
$Y \leftarrow \{y_{start}\}$;
```
/* main loop                                                   */
```
**for** $t = 1, \ldots, T$ **do**
    ```/* sample random element from the population            */```
    $\boldsymbol{\theta}_t \sim \mathcal{U}(P_{t-1})$;
    ```/* mutate                                               */```
    $d \sim \mathcal{U}(0, |\boldsymbol{\theta}_t|)$;              ```// sample random dimension```
    $\hat{\boldsymbol{\theta}} \leftarrow copy(\boldsymbol{\theta}_t)$;
    $\hat{\boldsymbol{\theta}}[d] \leftarrow \mathcal{U}(\Theta_d)$;     ```// sample a new value from the search space```
    ```/* evaluate                                             */```
    $y_t = [f_0(\hat{\boldsymbol{\theta}}), f_1(\hat{\boldsymbol{\theta}})]$;
    $Y \leftarrow Y \cup y_t$
    ```/* update population                                    */```
    $S(Y) = \{y' \in Y : \{y'' \in Y : y'' \succ y', y' \neq y''\} = \emptyset\}$;   ```// Pareto front```
    $P_t \leftarrow \{\boldsymbol{\theta} : y(\boldsymbol{\theta}) \in S(Y)\}$;
**end**

**Algorithm 5:** Local Search

