# OpenReview forum: "Structural Pruning of Large Language Models via Neural Architecture Search"
_ICLR.cc/2024/Conference — Submitted to ICLR 2024_

### Official Review · Reviewer_aZ9T · 2023-10-27

**Soundness:** 3 good
**Presentation:** 2 fair
**Contribution:** 2 fair
**Rating:** 3
**Confidence:** 4

**Summary:**

The paper suggests integrating weight-sharing NAS for the compression of pre-trained language models. This approach consists of three components: a weight-sharing super-network trained using the sandwich rule in conjunction with an in-place knowledge distillation (KD) strategy; a sub-network selection based on the Pareto front; and a varied search space, extending from a larger scope with masks applied to each head/neuron to a smaller scale focusing solely on the quantity of heads, units, and layers.

**Strengths:**

1. **NAS's Role in Compressing BERT**: A practical application for NAS is in the compression of BERT.
2. **Method Advantages for Sub-Network Selection**: This technique facilitates a multi-objective search for choosing multiple sub-networks, unlike earlier methods which allowed only single-network pruning and selection at a time.

**Weaknesses:**

1. **No experiments on LLM, but the topic of this paper is about LLM**. The title of the paper suggests a focus on Large Language Models , leading me to expect analyses or experiments involving LLMs like LLaMA, or at least T5-large, especially since the terms 'LLM' are predominantly used in the paper rather than 'language model' or 'pre-trained language model'. However, upon delving into the experimental section, it's surprising to find that the actual experiments exclusively involve **BERT**. There is no mention of LLMs in the experiment, nor is there any comparison or discussion on how the application of Neural Architecture Search might differ between LLMs and PLMs.

2. **More Baselines are needed**. The authors appear to have selected some baselines that may be relatively less challenging to outperform, such as Retraining-Free Pruning and self-defined baselines. Retraining-Free Pruning, focusing on rapid compression of BERT without retraining, isn't directly comparable with the methods in this paper, which involve around 12 hours of training (as indicated by 50,000 seconds in Figure 5) on the MNLI dataset. Moreover, the proposed method underperforms DistillBERT in more than half of the datasets (including MRPC, COLA, SST2, QNLI, MNLI). Other baseline methods are self-created, and there's a notable omission of any recent advancements in structural pruning methods for BERT in the past five years (e.g., DynaBERT, CoFi). Comparison with those methods are needed. Additionally, the related work section only references four papers on structural pruning for BERT, whereas the authors could have expanded their literature scope by referring to a broader survey[1] on this topic.

3. **Missing Important Comparison**. The paper misses an essential comparison with another study[2] that also employs NAS for compressing BERT. Given the relevance and slight methodological variation (NAS applied to pre-trained versus fine-tuned BERT) between the two studies, a comparison seems crucial. Both papers aim to achieve compression in downstream tasks, yet this paper lacks experimental evidence or analysis showing whether its method offers any advantages over the other one. This comparison is particularly pertinent since the approach to compression, whether on a pre-trained or fine-tuned model, could lead to different outcomes, and their exploration is essential for a comprehensive understanding.

[1] Survey on Model Compression and Acceleration for Pretrained Language Models.
[2] NAS-BERT: Task-Agnostic and Adaptive-Size BERT Compression with Neural Architecture Search

**Questions:**

Could you clarify the process used to calculate the runtime for each technique as shown in Figure 5? The RFP (Retraining-Free Pruning) paper mentions that pruning takes merely 0.01 hour, approximately 36 seconds. However, in your study, this duration extends to about 20,000 seconds. What factors contribute to this substantial discrepancy in runtime measurements?

---

> ### Author Response · Authors · 2023-11-21
> **rebuttal**
>
> We thank the reviewer for the constructive feedback.
>
> 1. To address their concerns, we updated our paper and replaced the term ‘large language model’ by ‘pre-trained language model’. Even though LLM with prompt-classification are increasingly popular, fine-tuning models such as BERT is still highly competitive on labeled datasets and often used in practice. We see the extension of our approach to LLM as a promising future work.
> 2. We tried to benchmark against COFI using the official implementation but obtained inconclusive results compared to the original paper. We decided not to include these results yet to avoid an unfair comparison that would put COFI at a disadvantage. We reached out to the authors to understand this in detail and ensure a fair comparison;  if possible, will included the results of experiments for the camera ready version. However, we would like to stress that including additional structural pruning methods would not change the main argument of the paper: NAS allows for multi-objective optimization rather than constrained optimization to obtain the Pareto set of sub-networks. This captures the non-linear relationship between performance and compression ratio. For example, Figure 5 right shows that pruning the model on the MNLI by 50% leads to the same performance as pruning by 60%. Since we do not know this before, we either have to run the constraint optimization process (e.g. COFI or RFP) multiple times based on a grid of threshold values or stick to a sub-optimal solution based on an arbitrarily defined threshold. Multi-objective NAS on the other hand allows us to select the model without any additional overhead.
> 3. Regarding the callout on comparison with distillation methods (NAS-BERT), we appreciate the feedback from the reviewer and included a more detailed discussion about the differences of distillation and pruning approaches in the related literature section. As we discuss in the related work section and also in the response to reviewer t4jT, in this setting, given a pre-trained model, the goal is to train a smaller model from scratch by distilling the knowledge of a larger pre-trained model. While distillation allows for flexible architectures, as is the case for NAS-BERT, its cost is in the same ballpark as pre-training the teacher model. In our case, we focus on pruning directly on the target task, which is the more appealing scenario for practitioners who do not have the compute to distill their own model and would rather fine-tune a pre-trained model. This makes it hard to perform a fair apple-to-apple comparison between these two approaches. In addition, the two approaches are complementary to each other, since we can always prune a distilled model.
> 4. Question regarding runtime: For all methods we compute the total runtime to obtain a Pareto set of solutions. This includes also fine-tuning the original model for RFP and running it multiple times with different thresholds.

---

### Official Review · Reviewer_t4jT · 2023-10-31

**Soundness:** 3 good
**Presentation:** 3 good
**Contribution:** 2 fair
**Rating:** 5
**Confidence:** 5

**Summary:**

The paper studys a important research direction for NAS: structural pruning using NAS. The paper explores weight-sharing based neural architecture search (NAS) as a form of structural pruning to find sub-parts of the fine-tuned network that optimally trade-off efficiency. Authors valicate the effectiveness of the proposed method for fine-tuned BERT models.

**Strengths:**

The writing of this paper is commendable as it is well-structured and easily comprehensible.  I believe that utilizing Neural Architecture Search (NAS) for pruning structured architectures is one of the crucial research directions in the field of NAS. This paper provides detailed experimental evidence of the effectiveness of their approach, particularly on the GLUE benchmark.

**Weaknesses:**

One significant aspect that requires attention is the performance on the GLUE benchmark. It is worth considering an alternative branch of NAS, which involves directly searching for new architectures using distillation techniques for fine-tuned models such as AdaBERT, TinyBERT, and NAS-BERT. These methods have demonstrated the ability to achieve 50% pruning without any notable performance degradation and even achieve an impressive 80% reduction in parameters with minimal impact on performance. It would be beneficial to include and discuss these baselines in the paper. Moreover, it would be interesting to explore the potential combination of the proposed methods with these existing models and highlight the advantages of the proposed approach. I believe that incorporating these insightful discussions would greatly enhance the paper.

**Questions:**

See weakness.

---

> ### Author Response · Authors · 2023-11-21
> **rebuttal**
>
> We thank the reviewer for the thoughtful feedback and we are glad that the reviewer appreciated the experimental evidence of the effectiveness of our approach. We completely agree that compressing pre-trained language models is a crucial research direction for NAS. Indeed distillation techniques such as AdaBERT and TinyBERT are interesting alternatives to utilize NAS. However, there are several reasons why an experiment against distillation approaches would not be an apple-to-apple comparison:
>
> * We tackle the multi-objective setting where we obtain a Pareto set as a solution. Distillation on the other hand only provides a single solution. Providing a Pareto set of solutions is one of the key contributions of our work.
> * Distillation requires an expensive offline phase to train a single student model, which can then be fine-tuned on target tasks. Our pruning approach does incur an additional overhead compared to conventional fine-tuning on the target task, but omits the expensive distilation phase; pruning is substantially faster than distillation. Both approaches are valid but it depends on how many downstream tasks are considers (# tuning per experiment) to decide which approach is more useful. This makes it hard to compare the overall required compute time of both approaches.
>
> As suggested by the reviewer, we added a detailed discussion of AdaBERT and TinyBERT to the related work section of our paper and expanded the discussion on differences between pruning and distillation methods. We already discuss NAS-BERT in our paper, which is orthogonal to our approach. Since our method is also independent of whether the network was pre-trained or distilled from another model, it is straightforward to combine it with methods such as NAS-BERT.

---

### Official Review · Reviewer_2Az8 · 2023-11-01

**Soundness:** 3 good
**Presentation:** 3 good
**Contribution:** 3 good
**Rating:** 6
**Confidence:** 2

**Summary:**

This paper proposed weight-sharing-based NAS to compress fine-tuned pre-trained LLMs by slicing subnetworks. By utilizing a multi-objective approach, they can find the Pareto optimal set of architectures that balance model size and validation error. The NAS approach achieves up to 50% compression with less than 5% performance drop for a fine-tuned BERT model on 7 out of 8 text classification tasks.

**Strengths:**

The paper has detailed literature research and multiple baseline models for comparison and the selected topic is very important given that LLM is more and more important in our everyday life. Improving the LLM efficiency is critical.

The paper also provided in-depth ablation study.

**Weaknesses:**

When looking the metrics, it seems the newly proposed NAS model mainly performs better when the model is pruned heavily. The model inference time is not the best when compared with other models.

**Questions:**

In Figure 4, the graph shows: On 7 out of 8 dataset the new NAS strategy is able to prune 50% with less than 5% drop in performance (indicated by the dashed line) in performance. It seems more than 1 datasets dropped more than 5% when pruning 50% of parameters?

---

> ### Author Response · Authors · 2023-11-21
> **rebuttal**
>
> We thank the reviewer for the thoughtful feedback and appreciate that they found the paper to be ‘very interesting’ and that they appreciate the in-depth ablation study of the paper.
>
> **Inference Time**: Could the reviewer clarify if they mean model inference time or the overall runtime of our method? We only report model inference time in Figure 11 in the appendix, where in fact the sub-networks found by our NAS approach are substantially faster than the quantized original network. Decreasing model inference time is one of the key benefits of our method and we use the network size as a proxy for infererence latency throughout the paper. We have added the following sentence in the results section to make it clear - note that parameter count is related to model inference time as discussed in Appendix.
>
> In the case of runtime, our weight-sharing based NAS approach indeed induces some additional time overhead compared to just fine-tuning the model. However, we would like to point out that this time is significantly smaller (see Figure 5 left) than standard NAS or dropping layers. Our multi-objective approach returns a set of solutions such that the optimal compressed network can be selected post-hoc. This is in contrast to other structural pruning approaches, such as layer dropping or RFP, which require a pre-defined threshold on the remaining parameters. Since there is no simple relationship between pruning ratio (i.e., number of parameters left in the model) and performance, to find the optimal sub-network, we have to run these methods multiple times with different thresholds. Having said that, we do think there is room for further research to make weight-sharing based NAS more time efficient, for example by using non-uniform sampling of sub-networks during the fine-tuning of the super-network.
>
> **Question Figure 4**: We’d like to ask the reviewer to clarify why it seems that more than 1 datasets dropped more than 5% when pruning 50% of parameters. Our experiments show that a performance drop higher than 5% was actually only observed on the COLA dataset.

---

### Author Response · Authors · 2023-11-21
**Rebuttal to all reviewers**

First of all, we thank all reviewers for taking the time to review our paper and providing constructive feedback. We appreciate that the reviewers found the topic of our paper ‘very important’ (2Az8) and that ‘improving LLM efficiency is critical' (2Az8). We fully agree with reviewer t4jT that 'utilizing Neural Architecture Search (NAS) for pruning structured architectures is one of the crucial research directions in the field of NAS’ (t4jT). We are also happy that reviewers found the writing of our paper 'commendable as it is well-structured and easily comprehensible' (t4jT).

We elaborate on more specific points in comments directly to reviewers. We updated our current draft (updates in red) and incorporated the feedback from the reviewers. In particular:

* As requested by reviewer aZ9T, we changed the term large language model to pre-trained language model to avoid any confusions.
* We extended the related work section to discuss the work mentioned by reviewer aZ9T and t4jT.

---

### Meta-Review · Area_Chair_4iEa · 2023-12-07

**Metareview:**

This paper proposes a weight-sharing based neural architecture search (NAS) approach for compressing large language models, focusing on fine-tuned BERT models. However, the paper has several significant weaknesses that warrant rejection. Firstly, the title and abstract suggest a focus on Large Language Models, but the experiments only involve BERT, with no analysis or experiments on LLMs like LLaMA or T5-large. Secondly, the paper lacks a comprehensive comparison with relevant baselines and recent advancements in structural pruning methods, leading to a less convincing evaluation. Lastly, the paper misses a crucial comparison with another study that also employs NAS for compressing BERT, which is essential for understanding the advantages of the proposed method. Due to these concerns, I recommend rejection of the paper in its current form.

**Justification For Why Not Higher Score:**

The paper's claim on LLMs is not quite a practical one as pointed out by the reviewers. The study is either not siginficant enough or do not have sufficient evidence to support a more exciting claim.

**Justification For Why Not Lower Score:**

n/a

---

### Decision · Program_Chairs · 2024-01-16

Reject